# Cost-Effectiveness, Efficacy, and Safety Analysis of Tailored Therapy in Patients with *Helicobacter pylori* Infection

**DOI:** 10.3390/jcm10122619

**Published:** 2021-06-14

**Authors:** A Reum Choe, Ki-Nam Shim, Yehyun Park, Eun-Mi Song, Chung Hyun Tae, Sung-Ae Jung

**Affiliations:** Department of Internal Medicine, College of Medicine, Ewha Womans University, Seoul 07985, Korea; archoi20@ewha.ac.kr (A.R.C.); splendidyh1029@ewha.ac.kr (Y.P.); rainbowtip@ewha.ac.kr (E.-M.S.); jhtae@ewha.ac.kr (C.H.T.); jassa@ewha.ac.kr (S.-A.J.)

**Keywords:** *Helicobacter pylori*, eradication, antibiotic resistance, tailored, empirical

## Abstract

Recently in Korea, where triple therapy is accepted as the first-line *Helicobacter pylori* (*H. pylori*) eradication treatment, antibiotic resistance to clarithromycin has increased considerably, resulting in eradication rates of less than 80%. We investigated the efficacy of tailored therapy after a clarithromycin resistance test compared with empirical therapy for *H. pylori* eradication. The cost-effectiveness of *H. pylori* eradication success was evaluated according to the average medical cost per patient. A total of 364 patients were enrolled in the study. The first-line *H. pylori* eradication rate was significantly higher in patients who received tailored therapy than in those who received empirical therapy. The total medical costs for the tailored and empirical groups were 46,374 Won and 53,528 Won. The total treatment period for each ultimately successful eradication in the tailored group was 79.8 ± 2.8 days, which is shorter than that of the empirical group (99.2 ± 7.4 days). The rate of eradication-related adverse events for the tailored group and empirical group was 12.9% and 14.8%, respectively. Tailored therapy could be a useful option to achieve a higher successful eradication rate, shorter treatment periods, and lower medical costs than empirical therapy in the era of increasing antibiotic resistance.

## 1. Introduction

*Helicobacter pylori* (*H. pylori*) is associated with peptic ulcer disease, mucosa-associated lymphoid tissue lymphoma, and gastric cancer [1,2,3], and Korean guidelines recommend triple therapy as the first-line eradication treatment for *H. pylori* infection [4]. However, due to the increase of antibiotic-resistant strains, the eradication rate has gradually decreased [5,6], and several clinical studies have reported that the eradication rate of triple therapy is less than 80% [7,8,9]. The causes of this overall decrease in eradication rate include antibiotic resistance, age, smoking status, the difference in host immunity, underlying disease, and poor drug compliance [10,11], but antibiotic resistance is known as the most important factor among them [12].

Kuo et al. recently reported on patients with refractory *H. pylori* infection in Taiwan. Dual resistance to both clarithromycin and levofloxacin was found in 73.2%. This study highlighted ways to decide the optimum *H. pylori* eradication strategy according to the results of antibacterial susceptibility analysis [13]. In Italy, current guidelines recommend 10-day bismuth-based or sequential and concomitant regimens for first-line *H. pylori* eradication. Bismuth-based and bismuth-free therapies are equally effective for first-line *H. pylori* eradication [14]. Therefore, in areas with high clarithromycin resistance, opinions have arisen that the eradication treatment should be improved by adding more antibiotics instead of the triple therapy as the primary treatment, or changing to a new antibiotic of another class [15].

An understanding of the mechanism of *H. pylori* resistance is extremely complex. The *H. pylori* virulence factors are involved in the induction of inflammatory responses, and control and regulate those responses, maintaining chronic inflammation [16]. The *H. pylori* exhibit an expanded complex of mechanisms that alter host cellular responses and signaling pathways. *H. pylori* elicit numerous adaptive mechanisms that enable effective bacterial adherence, colonization, and cellular alterations that provide the induction of further premalignant changes in the gastric microenvironment [17].

In the Maastricht V/Florence guidelines, bismuth-containing quadruple therapy or concomitant therapy is recommended in areas where the clarithromycin resistance rate is higher than 15% [18]. Recently, the clarithromycin resistance rate in Korea has been about 30% [19]. Based on the aforementioned guidelines, it is necessary to establish a treatment plan based on the results of antibiotic susceptibility tests rather than maintaining the existing standard triple therapy that is based on clarithromycin as the primary treatment. Resistance to clarithromycin is mostly caused by a point mutation at position 2142 or 2143 of the 23S ribosomal RNA gene [20], and antibiotic resistance can also be predicted by using a dual-priming oligonucleotide (DPO)-based multiplex polymerase chain reaction (PCR) test [21]. This method is an excellent test that shows a relatively high sensitivity of 82% to 90% and a specificity of 95% or more. Moreover, it has the advantage of being able to confirm the resistance to clarithromycin in addition to other antibiotics [22]. Therefore, it is possible to check the individual’s resistance to antibiotics and then perform tailored therapy for *H. pylori*.

Tailored therapy based on the DPO-PCR test could be a useful regimen to increase the eradication rate of *H. pylori* infection. However, the cost-effectiveness of this test has not yet been definitively identified. The DPO-based multiplex PCR is more expensive than the Giemsa stain and rapid urease tests. It is difficult to perform this test routinely in clinical practice before the eradication of *H. pylori*. Therefore, we aimed to evaluate the eradication rate, adverse drug events, and cost-effectiveness of empirical and tailored therapies for the treatment of *H. pylori* infection.

## 2. Materials and Methods

### 2.1. Study Design and Subjects

This retrospective study was performed at two university hospitals in South Korea (Ewha Womans University Mokdong Hospital and Ewha Womans University Seoul Hospital) from January 2017 to December 2019. We included 435 subjects who met the following criteria: (1) the presence of *H. pylori* was confirmed by rapid urease test, histology such as Giemsa stain, urea breath test (UBT), or DPO-PCR test; (2) patients receiving empirical therapy or tailored therapy based on the DPO-PCR results. Subjects (*n* = 71) were excluded in the following conditions: (1) patients aged >80 years (*n* = 13); (2) history of gastrectomy (*n* = 18); (3) severe systemic illness, such as severe cardiopulmonary dysfunction, liver cirrhosis, or renal failure (*n* = 17); (4) history of any allergic reaction to antibiotics (*n* = 8); (5) loss to follow-up (*n* = 15).

A total of 364 patients who received *H. pylori* eradication treatment (*n* = 121 in Ewha Womans University Mokdong Hospital and *n* = 243 in Ewha Womans University Seoul Hospital) were enrolled in this study. Data for January 2017 and January 2018 were collected through chart review in Ewha Womans University Mokdong Hospital, and data for February 2018 and December 2019 were collected using data extracted from the Clinical Data Warehouse of the Ewha Womans University Seoul Hospital. This study was approved by the Institutional Review Board (IRB approval number: 2020-09-013).

### 2.2. H. Pylori Diagnosis and DPO-Based Multiplex PCR

Infection with *H. pylori* was regarded as positive when at least one positive result was obtained in UBT (Otsuka^®^, Tokyo, Japan), rapid urease test (CLOtest^®^; Delta West, Bentley, Australia), or histologic assessment (Giemsa staining) conducted using gastric biopsy specimens from the antrum and greater curvature of the body. In the tailored therapy group, DNA was extracted from frozen gastric biopsy specimens to detect clarithromycin-resistant *H. pylori* mutants. DPO-based multiplex PCR (Seeplex^®^
*H. pylori*-ClaR ACE Detection; Seegene, Inc., Seoul, Korea) was performed. Point mutations were identified by PCR amplification of a portion of the 23S ribosomal RNA gene. The amplified DNA products were visualized on a UV transilluminator after electrophoresis on a 2% agarose gel. The amplified DNA products were determined to have point mutations.

### 2.3. H. Pylori Eradication Therapy Regimen

There were a total of three regimens for empirical therapy, which depended on the doctor’s preference: (1) triple therapy (proton-pump inhibitor bid, amoxicillin 1 g bid, clarithromycin 500 mg bid) for 14 days; (2) sequential therapy (proton-pump inhibitor bid and amoxicillin 1 g for 5 days followed by proton-pump inhibitor bid, clarithromycin 500 mg, and metronidazole 500 mg for 5 days) for 10 days; (3) bismuth-containing quadruple therapy (proton-pump inhibitor bid, bismuth subcitrate 300 mg qid, metronidazole 500 mg tid, tetracycline 500 mg qid) for 14 days. In the tailored therapy group, patients received the eradication regimen based on the results of the DPO-PCR test. Six weeks after the end of the treatment, compliance with therapy and side effects were assessed through personal interviews. An adverse event was defined as an unscheduled early visit to the outpatient clinic with symptoms during eradication or when the *H. pylori* eradication treatment was stopped due to adverse drug effects. UBT, rapid urease test, or Giemsa staining of gastric biopsy specimens were performed to confirm successful *H. pylori* eradication.

### 2.4. Medical Cost

The total medical cost per patient was assessed as the sum of the diagnostic and regimen costs. The cost of the DPO-PCR test was 38,350 Won, while those of the Giemsa stain and rapid urease test were 11,427 Won and 10,504 Won, respectively. The cost of 14-day triple therapy was calculated to be 34,300 Won. The cost of 10 day sequential therapy was 41,800 Won and that of the 14-day quadruple therapy was 25,746 Won.

### 2.5. Statistical Analysis

Continuous variables were presented as mean ± standard deviation, and categorical variables were presented as the number of subjects and percentage. Group comparisons were performed using independent samples *t*-tests or Mann–Whitney U tests for continuous variables and Pearson’s chi-squared tests or Fisher’s exact tests for categorical variables. Categorical variables are presented as numbers and proportions. All statistical analyses were 2-sided, and results were considered statistically significant at *p* < 0.05. The Statistical Package for the Social Science (SPSS) software (version 21.0; SPSS Inc., Chicago, IL, U.S.) was used for statistical analysis.

## 3. Results

### 3.1. Baseline Characteristics of Study Subjects

During the study period, 435 patients received *H. pylori* eradication treatment. After excluding 71 patients, 364 patients were finally enrolled in the study. Empirical therapy was given to 155 patients, and 209 patients received tailored therapy. There were older and more current smokers in the tailored therapy group. The most common cause of need for *H. pylori* eradication was *H. pylori* associated gastritis in both groups (Table 1).

### 3.2. H. pylori Eradication Rate of Study Subjects

Of 155 patients in the empirical therapy group, 111 (71.6%) received sequential therapy as the first-line *H. pylori* eradication regimen. Of 209 patients in the tailored therapy group, 133 (66.5%) received 7-day triple therapy as the first-line *H. pylori* eradication regimen. As for the eradication rate according to regimen, the 14-day triple therapy was 81.5%, bismuth-containing quadruple therapy was 88.2%, and sequential therapy was 82.0% in the empirical therapy group. In the tailored therapy group, the eradication rate for 7-day triple therapy was 89.2%, bismuth-containing quadruple therapy was 91.7%, and sequential therapy was 80.0%. The eradication rate was significantly higher in those who received tailored therapy than in those who received empirical therapy (82.6% vs. 91.2%; *p* = 0.023) (Table 2).

### 3.3. Medical Cost of Study Subjects

The total medical costs per tailored group were 46,374 Won and those for the empirical group were 53,528 Won (*p* < 0.001). The cost of the diagnostic method was higher in the tailored therapy group (22,914 Won vs. 11,282 Won; *p* < 0.001). The cost of the eradication regimen was higher in the empirical group (38,746 Won vs. 20,704 Won; *p* < 0.001). The total treatment duration for each ultimately successful eradication in the tailored therapy group was 79.8 ± 2.8 days, which is significantly shorter than that of the empirical group’s 99.2 ± 7.4 days (*p* = 0.013). (Figure 1)

### 3.4. Adverse Effects Seen in Study Subjects

Adverse drug events were more common in the empirical therapy group than in the tailored therapy group (14.8% vs. 12.9%, *p* = 0.028). The most common adverse event was nausea or vomiting in both groups (26.1% vs. 29.7%) (Table 3). Five patients in the empirical and three patients in the tailored therapy group discontinued *H. pylori* eradication treatment due to side effects. After *H. pylori* eradication failure, there were 6 patients (3.9%) in the empirical therapy group who did not want the next eradication treatment, but none of the patients in the tailored therapy group denied further treatment.

## 4. Discussion

In this study, the eradication rate of *H. pylori* was significantly higher in those who received tailored therapy as their first-line treatment in comparison to those who received empirical therapy. The *H. pylori* eradication regimen is more cost-effective in tailored therapy than empirical therapy. The total treatment periods for ultimately successful eradication were shorter in the tailored therapy group, and fewer eradication-related adverse drug events were observed compared to the empirical therapy group.

The first-line *H. pylori* eradication rate was significantly higher with tailored therapy than with empirical therapy. Recently, as the clarithromycin resistance rate of *H. pylori* has increased, the eradication rate of the existing triple therapy for seven days has decreased. To overcome this situation, prolonged treatment periods, various regimens such as quadruple therapy with bismuth, sequential and concurrent therapy, or tailored therapy may be used [8,23,24,25]. It is important to succeed as a first-line eradication treatment because the *H. pylori* eradication regimen requires a large amount of medicine, including two antibiotics to be taken. In particular, if you are already taking other drugs due to a comorbid disease, the *H. pylori* eradication treatment might a great burden. Moreover, antibiotic resistance is the leading cause of failure of the *H. pylori* eradication treatment, and we should try to reduce antibiotic resistance. Empirical quadruple therapy with bismuth as the first-line therapy raises public health concerns regarding increasing resistance to the constituent antibiotics. Finally, in tailored therapy, the risk of misuse of the antibiotics is lower due to reduced *H. pylori* eradication retreatment.

The *H. pylori* eradication regimen was more cost-effective in the tailored therapy than in the empirical therapy in this study. Different studies that have evaluated *H. pylori* tailored therapy have achieved contradictory results. Liou et al. in Taiwan found that 6920 USD would be required to additionally cure one patient using the genotype resistance guide therapy, compared to empirical therapy, which is clearly not a cost-effective option [26]. Chang et al. in South Korea evaluated the cost-effectiveness of tailored therapy, and compared the results of standard triple therapy with those of empirical bismuth quadruple therapy. Total per capita medical costs were 503.50 USD in the tailored group and 406.50 USD in the empirical group [27]. However, Cosme et al. reported that in Spain, the culture-based approach was more cost-effective than standard first-line therapy given empirically [28]. Gweon et al. showed that the cost for a successful eradication using DPO-based PCR would be similar or superior to the expected cost of a successful eradication with a 14-day empirical treatment when the first-line eradication rate is ≤80% [29]. Since *H. pylori* antibiotic resistance varies among different geographic areas, the cost-effectiveness may vary according to the cost of care in a given country, and therefore the same conclusion may not be applicable to other healthcare systems [30].

A novel view of *H. pylori* infections is emerging in microbiological point. The changes of gut microbiota are greatly implicated in the pathogenicity of *H. pylori* Infections. The antimicrobial peptides (AMPs) have great importance in the innate immune reactions to *H. pylori* and participate in conservative co-evolution with an intricate microbiome. During *H. pylori* infections, AMP expression is able to eradicate the bacteria, thereby preventing *H. pylori* infections in the gastrointestinal tract [31]. The β-Defensins which belong to the AMP group expression, are enhanced during *H. pylori* infection [32].

In chronic inflammation induced by *H. pylori* infection, COX-2 is modulated by DNA methylation. The DNA methylation changes at the COX-2 promoter are associated with transcriptional activation and precede histone modifications in gastric cells exposed to *H. pylori* [33]. Woo et al. reported that the genome-wide methylation profiles associated with *H. pylori* infection. The gastric cancer is regulated by methylation mechanism rather than genetic linkage, and *H. pylori* leads DNA methylation [34]. Therefore, methylation-based biomarkers could be used for monitoring the prognosis of treatment, drug response, and recurrence in gastric cancer.

The prevalence of 23S rRNA point mutations was 28.7% in the study population. A total of 139 patients (66.5%) who received 7-day triple therapy were included in the tailored therapy group. This regimen might decrease drug-related adverse events and result in shorter treatment duration and minimal antibiotic overuse. Better compliance and fewer adverse effects were observed in patients in the tailored therapy group, similar to those reported in a previous study [35,36]. Choi et al. showed that the rate of eradication–related side effects for tailored regimens was 12.0%, which differed significantly from that of empirical bismuth quadruple therapy for first-line *H. pylori* treatment [36]. Therefore, tailored therapy with DPO-based multiplex PCR for *H. pylori* eradication may be superior in quality, with fewer adverse events, compared to empirical therapy in Korea, where clarithromycin resistance is high.

There are several limitations in this study. First, patients with a negative *H. pylori* infection were not enrolled in this study, and the total medical cost would be underestimated. Second, as this study was conducted retrospectively, there were limitations in obtaining detailed medical information that could have an influence on eradication failure or diagnosis of infection. Third, this study enrolled patients from only two medical centers, and a majority of them resided in Seoul. This study may have been subject to a selection bias. Fourth, DPO-based multiplex PCR could determine only the presence of clarithromycin resistance, and we did not check for resistance to other antibiotics such as A2115G, G2141A, A2142T, and T2182C.

## 5. Conclusions

In conclusion, this study showed that tailored therapy could be a useful option to achieve a higher successful eradication rate, shorter treatment periods, and lower medical costs than empirical therapy in the era of increasing antibiotic resistance.

## Figures and Tables

**Figure 1 jcm-10-02619-f001:**
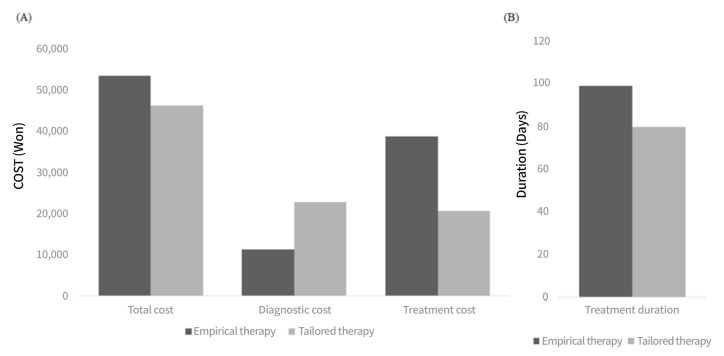
The total medical cost and treatment duration in patients on tailored therapy vs. empirical therapy (**A**) The total medical cost consisting of diagnostic and treatment cost in patients treated with tailored and empirical therapy. (**B**) The duration of treatment for each ultimately successful eradication in patients treated with tailored and empirical therapy.

**Table 1 jcm-10-02619-t001:** Baseline characteristics of empirical and tailored therapy groups.

	Total (*n* = 364)	Empirical Therapy (*n* = 155)	Tailored Therapy (*n* = 209)	*p* Value
Age (years), mean ± SD	56.0 ± 12.5	54.8 ± 11.7	56.9 ± 13.0	0.117
Sex, *n* (%)				
Male	197 (54.1)	79 (51.0)	118 (56.5)	0.298
Female	167 (45.9)	76 (49.0)	91 (43.5)	
Smoking, *n* (%)	108 (29.7)	37 (23.9)	71 (34.0)	0.037
Alcohol drinking, *n* (%)	110 (30.2)	56 (36.1)	54 (25.8)	0.034
Disease for *H. pylori* eradication, *n* (%)				<0.001
Peptic ulcer	109 (29.9)	39 (25.2)	70 (33.5)	
Post-ESD for EGC	24 (6.6)	5 (3.2)	19 (9.1)	
Post-ER for adenoma	35 (9.6)	11 (7.1)	24 (11.5)	
MALT Lymphoma	4 (1.1)	2 (1.3)	2 (1.0)	
*H. pylori* gastritis	175 (48.1)	93 (60.0)	82 (39.2)	
Lymphoid follicular gastritis	13 (3.6)	4 (2.6)	9 (4.3)	
Family history of gastric cancer	4 (1.1)	1 (0.6)	3 (1.5)	

Abbreviations: SD, standard deviation; ESD, endoscopic submucosal dissection; EGC, early gastric cancer; ER, endoscopic resection; MALT lymphoma, mucosa-associated lymphoid tissue lymphoma.

**Table 2 jcm-10-02619-t002:** *H. pylori* eradication rate of study subjects.

		Empirical Therapy (*n* = 155)	Tailored Therapy (*n* = 209)	*p* Value
1st line eradication regimen, *n* (%)	Triple	27 (17.4)	139 (66.5)	-
	Quadruple	17 (11.0)	60 (28.7)	-
	Sequential	111 (71.6)	10 (4.8)	-
1st line eradication rate according to regimen, *n* (%)	Triple	22/27 (81.5)	124/139 (89.2)	0.328
	Quadruple	15/17 (88.2)	55/60 (91.7)	0.646
	Sequential	91/111 (82.0)	8/10 (80.0)	1.000
Outcome of *H. pylori* 1st line eradication, % (*n*/*N*)				
Eradication rate in analysis		82.6 (128/155)	91.7 (187/204)	0.023

**Table 3 jcm-10-02619-t003:** *H. pylori* eradication related adverse effects of study subjects.

	Empirical Therapy	Tailored Therapy	*p* Value
Adverse event for eradication treatment, *n* (%)	23 (14.8)	27 (12.9)	0.028
Abdominal pain	5 (21.8)	2 (7.4)	
Nausea/Vomiting	6 (26.1)	8 (29.7)	
Headache	4 (17.4)	2 (7.4)	
Diarrhea	5 (21.7)	4 (14.8)	
Dyspepsia	0 (0.0)	7 (25.9)	
Metallic taste	3 (13.0)	4 (14.8)	
No further treatment after eradication fail, *n* (%)	6 (3.9)	0 (0.0)	NA

## Data Availability

The datasets generated and/or analyzed during the current study are not publicly available due to our IRB policy but are available from the corresponding author upon reasonable request.

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
