# Peer review of "Cost-Effectiveness, Efficacy, and Safety Analysis of Tailored Therapy in Patients with *Helicobacter pylori* Infection"

_jcm, 2021, doi:10.3390/jcm10122619_

Round 1

Reviewer 1 Report

The paper entitled: “Cost-effectiveness, Efficacy, and Safety Analysis of Tailored Therapy in Patients with Helicobacter pylori Infection”, is well structured.

However, I have some suggestions for the authors:

  • All paragraphs must have the same setting, read the newspaper guidelines carefully.
  • The figures must be of higher resolution, they are faded, in particular the diagram is not very legible.
  • In addition to seeing the effects on a physiological level (naurea, abdominal pain), the authors should study what happens at a molecular level.
  • In the Discussion in line 204 where it is named, Cosme et al., I suggest the authors to broaden the discussion by emphasizing the importance of antimicrobial activity (Pero et al DOI: 10.3390 / biom9060237).
  • Also in the discussion it should be added whether these results can be the springboard for future methylation studies (Angrisano et al DOI: 10.1371 / journal.pone.0156671)

Reviewer 2 Report

Dear Editor,

I have read with interest the study by Choe et al. This is a retrospective study of patients enrolled in two Korean centers in Seoul and Mokdong (near Seoul). The Authors want to argue the advantages of a tailored Helicobacter Pylori therapy over empirical therapies. The study immediately describes the increase in resistant clarithromycin patients in Korea (page1, line 37-40), bringing the percentage of resistant to about 30%. In this case, as correctly pointed out, the guidelines require the use of quadruple therapy with bismuth. The purpose of the study is clearly described, as are the limitations.

Introduction -  consider including references:

  • Kuo CJ et al Multidrug resistance: The clinical dilemma of refractory Helicobacter pylori infection. Microbiol Immunol Infect. 2021 Mar 23:S1684-1182(21)00059-1. doi: 10.1016/j.jmii.2021.03.006. (on page 1, line 33)
  • De Francesco V et al. Quadruple, sequential, and concomitant first-line therapies for H. pylori eradication: a prospective, randomized study. Dig Liver Dis. 2018 Feb; 50 (2): 139-141. doi: 10.1016 / j.dld.2017.10.009. (on page 1, line 36).

The authors enrolled 364 patients from 569 Hp positive patients (page 2, line 65). However, 71 patients were excluded (page 2, line 75). But 569-71 = 498. In the results, 435 patients received the eradicating treatment. This means that 134 patients did not receive the treatment. This description of patient enrollment / exclusion may not be clear.

Figure 1 is redundant (especially for patients excluded in accordance with the exclusion criteria already described in the text on page 2 line 75). Overall, this figure does not help in understanding the study.

The next paragraph (page 4, line 132-137) is also redundant because it is clearly represented in Table 1.

Page 6: I would move the reference to Table 3 from line 173 to line 170, at the end of the sentence "The most common advers event ...".

It is unclear how many patients discontinued treatment due to side effects.

In conclusion, I believe that this study is economically interesting and that it rightly points out that empirical therapies should be less and less adopted than tailored. 

My opinion is that this study is going in the right direction. However, the major limitation is that, as the authors themselves describe, Korea is now a country that has far exceeded the 15% quota of patients resistant to clarithromycin. This, paradoxically, would mean that the empirical scheme to be adopted should only be that dedicated to clarithromycin-resistant (bismuth-containing quadruple therapy or concomitant). In fact, with 30% of patients positive, the DPO-PCR test would no longer have reason to be performed. This situation could quickly make this study out of date with clinical routine.

Reviewer 3 Report

Dear Authors,

H. pylori infection is associated with far more diseases/impairments so I would recommend correcting the very first sentence in the introduction. Besides, I would mention American College of Gastroenterology (ACG) Clinical Guideline rather than Korean recommendations. 

H. pylori should be written everywhere in italics. Please correct it in the whole text. 

Besides, the mechanism of bacterial resistance is quite complex and could be more evaluated. Please check https://doi.org/10.3390/cells10010027 

I would recommend improve the quality of Figure 1 since in this form it is quite blurred.

Line 198 - what does $ has to mean?

Round 2

Reviewer 2 Report

I appreciated the changes made by the authors and I am satisfied with the responses sent.